# Compositional Neuro-Symbolic Reasoning

## Abstract

We study structured abstraction-based reasoning for the Abstraction and Reasoning Corpus (ARC) and compare its generalization to test-time approaches. Purely neural architectures lack reliable combinatorial generalization, while strictly symbolic systems struggle with perceptual grounding. We therefore propose a neuro-symbolic architecture that extracts object-level structure from grids, uses neural priors to propose candidate transformations from a fixed domain-specific language (DSL) of atomic patterns, and filters hypotheses using cross-example consistency checks. Instantiated as a compositional reasoning framework based on unit patterns inspired by human visual abstraction, the system augments large language models (LLMs) with object representations and transformation proposals. On ARC-AGI-2, it improves base LLM performance from 16% to 24.4% on the public evaluation set, and to 30.8% when combined with ARC Lang Solver via a meta-classifier[1]. These results support the claim that in the evaluated settings, allocating test-time compute to object abstraction, neural-guided transformation proposal, and consistency filtering can improve accuracy over unstructured sampling-based scaling without task-specific finetuning of the system.

## 1 Introduction

The Abstraction and Reasoning Corpus (ARC) was introduced as a benchmark for measuring fluid intelligence: the ability to infer abstract transformation rules from a small number of examples and apply them to novel inputs (Chollet, 2019; Lake et al., 2017). ARC-AGI-2 extends this paradigm by curating tasks that emphasize multi-step compositional reasoning, context-dependent transformations, and systematic generalization under minimal supervision (ARC Prize Foundation, 2025a). Each task provides only a handful of input–output grid pairs, and success requires identifying a latent rule that is consistent across all examples and generalizes to an unseen test instance. Concretely, the model observes colored-grid demonstrations, infers operations such as moving, recoloring, filling, or composing objects, and must produce an exact output grid for a held-out test input; a prediction is correct only if every cell matches the target. ARC-AGI is a grid-to-grid synthesis task rather than a classification or multiple-choice task: the system must generate the entire output grid for a new input, and a single incorrect cell makes the prediction wrong. Unlike conventional supervised learning benchmarks, ARC-AGI-2 offers no training distribution to exploit and explicitly limits the effectiveness of brute-force enumeration and memorization (Marcus, 2018; Geirhos et al., 2020).

Despite rapid advances in large language models (LLMs) and test-time scaling (Brown et al., 2020; Kaplan et al., 2020), ARC-AGI-2 continues to expose fundamental weaknesses in current reasoning architectures. End-to-end neural models entangle perception and rule induction, often producing brittle extrapolations when faced with novel compositions (Battaglia et al., 2018). Symbolic program synthesis methods incur a combinatorial explosion when searching over high-resolution grids and transformations (Gulwani, 2011). LLM-based solvers mitigate search through extensive sampling and self-consistency (Wei et al., 2022; Wang et al., 2023), but rely on probabilistic aggregation rather than explicit cross-example consistency checks, resulting in high computational cost and unstable generalization (Valmeekam et al., 2023).

We posit that ARC-AGI-2 is best approached by explicitly separating perceptual abstraction from rule induction and by constraining reasoning to a compact, reusable set of atomic visual transformations (Andreas et al., 2016; d'Avila Garcez et al., 2019). Under this view, the central problem is identifying a small set of

---

[1]At the time of release (November 2025), our approach achieved state-of-the-art performance under this experimental setup.

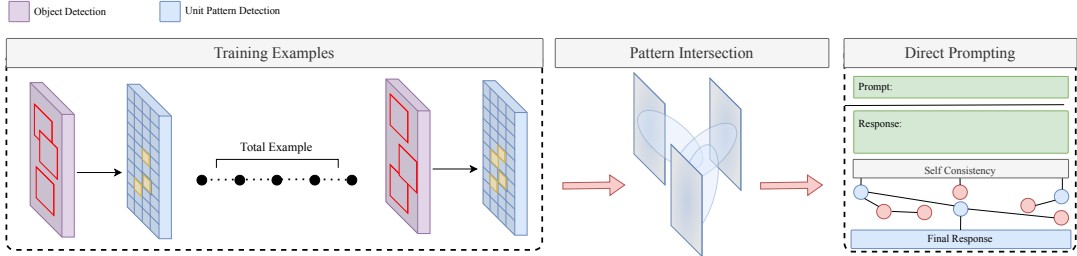

Figure 1: The neuro-symbolic pipeline first extracts object-level representations, including connected components and structured attributes. A neural prior proposes transformations from a constrained DSL. Candidate hypotheses are then checked for cross-example agreement and forwarded to test-time solution generation.

symbolic operations that jointly explain all training examples and compose coherently on the test input. This requires grounded object representations, guided hypothesis generation, and global consistency checking.

We instantiate this idea as a compositional neuro-symbolic architecture for ARC-AGI-2. Our framework extracts object-level structure from input grids, uses neural priors to propose candidate transformations from a fixed domain-specific language (DSL), and filters hypotheses through cross-example agreement. By decoupling perception, transformation proposal, and consistency verification, the system restricts combinatorial search while preserving flexibility for multi-step and context-sensitive reasoning. Empirically, the system achieves 24.4% on the ARC-AGI-2 public evaluation set, and when combined with ARC Lang Solver via a meta-classifier, reaches 30.8%. These results indicate that object-level abstraction, neural-guided hypothesis generation, and structured constraint checks improve performance in the evaluated setting without task-specific finetuning of the system, while reducing reliance on brute-force search and test-time scaling.

## 2 Methodology: ARC-AGI Compositional Reasoning

Our architecture implements a four-stage neuro-symbolic pipeline grounded in a single design principle: explicit separation between perceptual abstraction and rule induction. Given an ARC task with $k$ training pairs

$$(I_i, O_i) \quad \text{for } i = 1, \ldots, k,$$

each $I_i, O_i \in \{0, \ldots, 9\}^{N_i \times M_i}$ is a discrete grid over the 10 ARC colors. Here, $N_i$ and $M_i$ denote grid height and width, respectively. The objective is to infer a transformation program $\mathcal{T}$ such that

$$\mathcal{T}(I_i) = O_i \quad \forall i \in \{1, \ldots, k\},$$

and generalize $\mathcal{T}$ to the unseen test input. The pipeline proceeds sequentially: (1) structured object abstraction, (2) neural-guided hypothesis proposal over a fixed DSL, (3) cross-example consistency filtering, and (4) guided solution generation for the test grid.

### 2.1 Stage 1: Structured Symbolic Scene Abstraction

Stage 1 maps a raw grid $I \in \{0, \ldots, 9\}^{N \times M}$ to a structured symbolic scene graph. In the deployed system, low-level attributes such as connected components, bounding boxes, centroids, and color histograms are computed algorithmically, while higher-level descriptors such as shape labels or cavity cues may be enriched by prompted LLM analyses when they simplify downstream reasoning. We therefore use the term *structured symbolic abstraction* rather than a purely parameter-free symbolic parser.

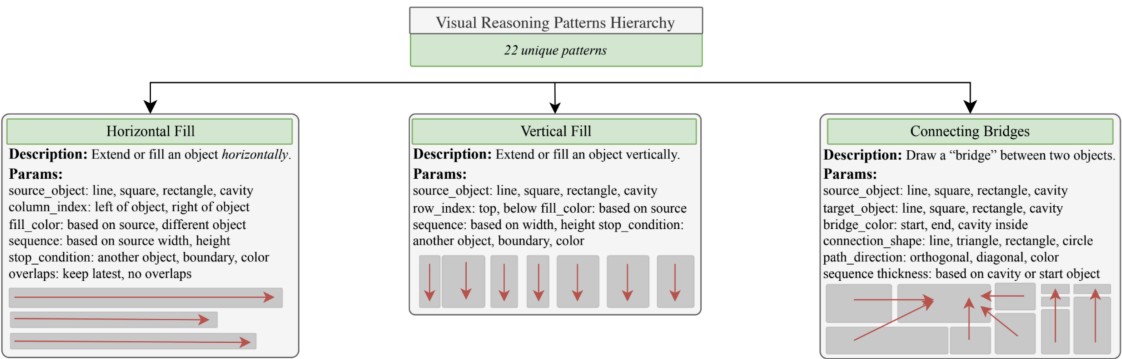

Figure 2: Hierarchy of atomic visual reasoning patterns used in compositional ARC solving. Each pattern corresponds to a primitive operation within a constrained DSL, parameterized by object attributes such as color, position, or connectivity. The hierarchy illustrates how complex transformations can be composed from a small set of reusable units, restricting combinatorial search.

### 2.1.1 Background Estimation

We define the background color as the mode of the grid:

$$c_{\text{bg}} = \arg \max_{c \in \{0, \ldots, 9\}} \sum_{y=1}^{N} \sum_{x=1}^{M} \mathbf{1}[I[y, x] = c].$$

Here, $\mathbf{1}[\cdot]$ denotes the indicator function, equal to 1 when the condition holds and 0 otherwise. The quantity inside the summation counts the number of pixels assigned color $c$. The background color $c_{\text{bg}}$ is thus the most frequent color in the grid. This definition ensures invariance to spatial arrangement and aligns with ARC's convention that background dominates in area. In implementation, this mode-based rule is the default heuristic, with prompted disambiguation used only for ambiguous cases.

### 2.1.2 Connected-Component Decomposition

Non-background pixels are partitioned into objects via 8-connected components. The adjacency relation

$$(y, x) \sim (y', x') \iff |y - y'| \leq 1 \ \wedge \ |x - x'| \leq 1.$$

For each pixel $(y, x)$ satisfying $I[y, x] \neq c_{\text{bg}}$ and not yet assigned to an object, we perform a Breadth-First Search (BFS) over the adjacency relation. The resulting object is

$$o_j = \{(y, x) \in \Omega : I[y, x] \neq c_{\text{bg}} \text{ and reachable via } \sim\},$$

where $\Omega = \{1, \ldots, N\} \times \{1, \ldots, M\}$ denotes the pixel index set. Each $o_j$ is therefore a maximal connected subset of non-background pixels. Deterministic scan order guarantees reproducibility.

### 2.1.3 Object Feature Parameterization

For each object $o_j$, we compute a structured feature representation $\phi(o_j)$.

**Bounding Box.**

$$y_{\min}^{(j)} = \min_{(y,x) \in o_j} y, \quad x_{\min}^{(j)} = \min_{(y,x) \in o_j} x,$$

$$y_{\max}^{(j)} = \max_{(y,x) \in o_j} y, \quad x_{\max}^{(j)} = \max_{(y,x) \in o_j} x.$$

These define the minimal axis-aligned rectangle enclosing $o_j$. The bounding box size is

$$h_j = y_{\max}^{(j)} - y_{\min}^{(j)} + 1, \quad w_j = x_{\max}^{(j)} - x_{\min}^{(j)} + 1,$$

where $h_j$ and $w_j$ denote object height and width.

**Centroid.**

$$(\bar{y}_j, \bar{x}_j) = \left( \frac{1}{|o_j|} \sum_{(y,x) \in o_j} y, \frac{1}{|o_j|} \sum_{(y,x) \in o_j} x \right).$$

Here, $|o_j|$ is the object's pixel count. The centroid encodes global position independent of shape.

**Canonical Shape Representation.** To remove translation variance, we define

$$\tilde{o}_j = \{(y - y_{\min}^{(j)}, x - x_{\min}^{(j)}) : (y,x) \in o_j\}.$$

This normalized coordinate set preserves geometry while fixing the origin at the bounding-box corner.

**Color Histogram.**

$$h_c(o_j) = \sum_{(y,x) \in o_j} \mathbf{1}[I[y,x] = c].$$

For each color $c$, $h_c(o_j)$ counts its frequency within the object, forming a 10-dimensional histogram vector.

**Cavity Detection.** Let $\mathcal{B}_j$ denote the bounding-box region of $o_j$. A cavity is a maximal connected set $B \subset \mathcal{B}_j$ such that

$$I[y,x] = c_{\mathrm{bg}} \quad \forall (y,x) \in B,$$

and

$$B \cap \partial \mathcal{B}_j = \varnothing,$$

where $\partial \mathcal{B}_j$ denotes the bounding-box boundary. This condition enforces full enclosure. Cavity count and geometry become part of $\phi(o_j)$.

### 2.1.4 Scene Graph Representation

The final symbolic abstraction of grid $I$ is

$$S(I) = \{o_1, \ldots, o_K\},$$

where each $o_j$ is parameterized by $\phi(o_j)$. $K$ denotes the number of detected objects. $S(I)$ is a structured symbolic scene graph used as the main input to neural-guided hypothesis generation.

## 2.2 Stage 2: Neural-Guided Hypothesis Generation

Given symbolic scene graphs $\{S(I_i), S(O_i)\}_{i=1}^k$, this stage proposes candidate transformations from a fixed DSL of 22 atomic Unit Patterns,

$$\mathcal{P} = \{p_1, \ldots, p_{22}\}.$$

The DSL is hybrid. Some primitives are deterministic grid or scene-graph operators. Others are LLM-assisted detectors that return a structured hypothesis

$$h = (r, \eta, \rho),$$

where $r$ indexes a Unit Pattern, $\eta$ contains its parameters, and $\rho$ is a short rationale. LLM-assisted hypotheses are not accepted as free-form text: they must satisfy a structured output schema and refer to valid objects, colors, positions, or relations extracted in Stage 1.

For deterministic primitives, a finite composition can be viewed as an executable program

$$\pi = p_{a_m} \circ p_{a_{m-1}} \circ \cdots \circ p_{a_1}.$$

This gives the idealized symbolic objective used when an explicit program can be instantiated. In the deployed system, however, the neural proposal distribution

$$q_\theta(h \mid S(I_i), S(O_i))$$

is realized by repeated structured LLM detections over the DSL, producing ranked Unit-Pattern hypotheses and parameterizations rather than an exhaustively enumerated program tree. The resulting hypothesis set for example $i$ is denoted $H_i$ and passed to Stage 3.

### 2.3 Stage 3: Cross-Example Consistency Filtering

This stage retains hypotheses that recur across demonstrations and remain compatible with the observed input-output pairs. For executable hypotheses, consistency is checked by rendering the predicted scene and comparing it with the target grid:

$$h \models i \iff \text{render}(\pi_h(S(I_i))) = O_i,$$

where $\pi_h$ is the instantiated executable program for hypothesis $h$. For LLM-assisted hypotheses without a complete executable program, we instead apply contract checks: schema validity, valid references to detected objects and colors, compatibility with the observed input-output differences, and recurrence across examples.

The ideal symbolic objective is to retain the intersection of hypotheses valid for every demonstration. Operationally, we approximate this objective by repeated structured pattern detection and consensus filtering: only Unit Patterns and parameterizations that pass the relevant checks and recur across examples are forwarded. When an explicit symbolic program can be instantiated, it is applied directly. Otherwise, the highest-ranked consensus hypotheses are converted into a compact structured hint and forwarded to Stage 4.

### 2.4 Stage 4: Guided Solution Generation for the Test Input

After identifying a consensus set of transformation hypotheses (Stage 3), the system constructs a structured hint for the unseen test input rather than assuming that every task admits a single symbolic program.

Let $I_{\text{test}}$ denote the test input grid and let $H_{\text{test}}$ denote the retained hint set obtained from recurrent Unit Patterns, their parameterizations, and the scene representation $S(I_{\text{test}})$. The final solver then predicts one or more candidate output grids according to

$$\widehat{O}_{\text{test}}^{(1)}, \ldots, \widehat{O}_{\text{test}}^{(N)} \sim r_\psi\big(\cdot \mid I_{\text{test}}, H_{\text{test}}, \{(I_i, O_i)\}_{i=1}^k\big),$$

where $r_\psi$ denotes a rule-based executor or an LLM conditioned on demonstrations and the structured hint.

When self-consistency is enabled, the $N$ candidate grids are aggregated by cell-wise majority voting to obtain the final prediction. In the ensemble setting, candidates are further combined with ARC Lang Solver (Berman, 2025) outputs through a meta-classifier. Stage 4 should be understood as guided solution synthesis from consensus structured hints rather than closed-form rendering of a single executable program.

## 3 Results

We evaluate our pipeline on the public evaluation set ARC-AGI-2 under the *pass@2* metric.

**Overall Performance.** Table 1 summarizes the results. The standalone Compositional Reasoner achieves **24.4%**, while the Meta-Classifier ensemble reaches **30.8%**. This places the method among the strongest

Table 1: **ARC-AGI-2 public evaluation performance under the official pass@2 metric.** A task is considered solved if at least one of the two submitted outputs exactly matches the ground-truth grid. The Meta-Classifier (ours) produces the two submitted outputs by running twice over a pool of four candidates generated by two independent pipelines, removing the first selected candidate before the second pass.

| System | Category | Score (%) |
|---|---|---|
| Human Panel | Human | 100.0 |
| **Meta-Classifier (Ours)** | Neuro-Symbolic + Ensemble | **30.8** |
| Compositional Reasoner (Ours) | Neuro-Symbolic | 24.4 |
| J. Berman | Hybrid | 29.4 |
| NVARC | Hybrid | 27.6 |
| GPT-5-Pro | LLM | 18.3 |
| Grok-4 (Thinking) | LLM | 16.0 |
| Claude Opus 4 (16K) | LLM | 8.6 |
| o3 (High) | LLM | 6.5 |
| o4-mini (High) | LLM | 6.1 |
| Claude Sonnet 4 (16K) | LLM | 5.9 |
| o3-Pro (High) | LLM | 4.9 |
| Gemini 2.5 Pro (32K) | LLM | 4.9 |

systems in our comparison. The improvement from 24.4% to 30.8% reflects the complementarity between structured compositional reasoning and LLM-based instruction synthesis. The meta-selection layer does not generate new solutions; it selects from an existing candidate pool. The observed gain, therefore, indicates that different pipelines solve partially overlapping but non-identical subsets of tasks.

**Comparison to Pure LLM Systems.** Under the matched-token comparison used for the LLM rows in Table 1, frontier LLMs without structured symbolic constraint, including chain-of-thought or extended-reasoning modes where available, achieve between 4.9% and 18.3%; the strongest pre-September 2025 baselines are GPT-5-Pro at 18.3%, Grok-4 (Thinking) at 16.0%, and Claude Opus 4 (16K) at 8.6%. The standalone Compositional Reasoner reaches 24.4%. We therefore interpret the gap as evidence that, on ARC-AGI-2, structured abstraction and DSL-guided hypothesis restriction can be a more effective use of test-time compute than unstructured stochastic frontier-model reasoning. This comparison directly addresses whether the observed gain can be explained only by additional test-time budget.

**Ablative Intuition: Where the Gains Arise.** The standalone Compositional Reasoner (24.4%) exceeds the matched LLM baselines in Table 1 under this evaluation setup. The additional +6.4 percentage points obtained by the Meta-Classifier indicate that different generators capture distinct transformation families. Tasks solved uniquely by the compositional pipeline typically involve cavity reasoning, structured fills, and symmetry constraints. Tasks solved uniquely by the LLM-based solver often involve higher-level semantic reinterpretation of object groupings. The ensemble does not rely on stochastic averaging; it operates as a discriminative selector under *pass@2* and reflects structural diversity rather than generation capacity.

**Remaining Gap to Human Performance.** Despite strong AI performance, the gap to 100% human accuracy remains substantial. Failure cases cluster around deeply compositional transformations requiring multi-stage relational reasoning or implicit latent grouping beyond the current 22-pattern DSL. These results suggest that while structured abstraction improves performance in the evaluated settings, further expansion of transformation primitives and global constraint solvers are required to approach human-level performance.

### 3.1 Ablation Studies

**Component Analysis of the Compositional Reasoner.** We analyze the contribution of the two core mechanisms in the Compositional Reasoner: (i) the *Symbolic Hint* pipeline (Stages 1–3), which constrains hypothesis space via object abstraction and cross-example pattern consensus, and (ii) the *Self-Consistency* (SC) decoding strategy in the final LLM solving stage. All experiments are conducted on the ARC-AGI-2 public evaluation set under the official *pass@2* metric. We evaluate four configurations obtained by selectively disabling components while keeping the underlying LLM fixed. The Full Model includes both symbolic hints

Table 2: **Ablation of the Compositional Reasoner on ARC-AGI-2 (pass@2).** $T_{\mathrm{sym}}$ and $C_{\mathrm{sym}}$ denote symbolic preprocessing latency and cost. $T_{\mathrm{llm}}$ and $C_{\mathrm{llm}}$ denote single LLM inference latency and cost. $N$ denotes the number of self-consistency samples. The highlighted row corresponds to the full proposed model.

| Configuration | Score (%) | Relative Latency | Relative Cost |
|---|---|---|---|
| **Full Model (Hints + SC)** | **24.4** | $T_{\mathrm{sym}} + NT_{\mathrm{llm}}$ | $C_{\mathrm{sym}} + NC_{\mathrm{llm}}$ |
| Hints Only (No SC) | 20.5 | $T_{\mathrm{sym}} + T_{\mathrm{llm}}$ | $C_{\mathrm{sym}} + C_{\mathrm{llm}}$ |
| SC Only (No Hints) | 17.5 | $NT_{\mathrm{llm}}$ | $NC_{\mathrm{llm}}$ |
| Baseline (Greedy LLM) | 15.0 | $T_{\mathrm{llm}}$ | $C_{\mathrm{llm}}$ |

Table 3: **Effect of meta-level candidate selection under pass@2.** All scores are measured on the ARC-AGI-2 public evaluation set. The highlighted row corresponds to the full proposed ensemble model.

| Solver Configuration | Score (%) |
|---|---|
| **Meta-Classifier Ensemble (Ours)** | **30.8** |
| ARC Lang Solver (Alone) | 26.6 |
| Compositional Reasoner (Alone) | 24.4 |

and SC voting. The Hints-only variant removes SC and performs single greedy decoding. The SC-only variant removes symbolic hints but retains $N$ stochastic samples with voting. The Baseline removes both contributions and performs single greedy decoding directly from raw examples. These ablations should be read as within-system component ablations rather than a complete compute-matched proof, since the full pipeline also includes structured preprocessing and additional context. Symbolic hints contribute the largest single drop when removed $(24.4 \to 17.5)$, while removing self-consistency reduces performance from $24.4\%$ to $20.5\%$. Thus, Table 2 supports the conclusion that structured hints are an important component of the pipeline, while matched-token leaderboard comparison in Table 1 addresses the compute-matching question.

**Meta-Classifier Ensemble Analysis.** The final system combines two independent solvers under the *pass@2* metric. The Compositional Reasoner generates two candidates, and ARC Lang Solver (Berman, 2025) is used as an external candidate generator that contributes two additional candidates. A meta-classifier selects one output at a time from the four-candidate pool, and the *pass@2* submission is formed by running this selector twice without replacement. Let $S_c$ denote the score of the Compositional Reasoner alone, $S_a$ the score of ARC Lang Solver alone, and $S_e$ the ensemble score after meta-selection. Empirically, the standalone Compositional Reasoner obtains $24.4\%$, while the ensemble improves over the strongest individual solver by 4.2 percentage points $(26.6 \to 30.8)$. This gain indicates that the two solvers capture partially complementary task families. The meta-classifier does not synthesize transformations; it operates as a discriminative selector.

**Impact of Individual Components on Accuracy.** Within the implemented pipeline, removing symbolic hints produces a larger drop than removing self-consistency (6.9 vs. 3.9 percentage points). We therefore treat symbolic hints as the dominant component in this system, while self-consistency provides additional robustness. This evidence is component-level rather than a full causal isolation of all sources of improvement, because preprocessing, prompt context, and call structure also change across configurations.

## 3.2 Computational Trade-offs

**Symbolic Pipeline Cost** $(T_{\mathrm{sym}}, C_{\mathrm{sym}})$**.** The symbolic stages incur a fixed additive cost per task. This includes structured object extraction and neural primitive detection. Because these operations are executed once per task, their cost scales additively rather than multiplying with the number of final solver samples.

**Self-Consistency Cost** $(NT_{\mathrm{llm}}, NC_{\mathrm{llm}})$**.** Self-consistency introduces a multiplicative cost factor of $N$, where $N$ is the number of sampled generations. Since LLM inference is repeated for each sample, this component becomes the source of multiplicative inference cost. Configurations employing self-consistency therefore, experience substantially higher latency compared to single-pass decoding. The trade-off structure is asymmetric: symbolic hints provide accuracy gains with additive rather than multiplicative cost.

**Efficient Operating Point.** The configuration without self-consistency (Hints Only) achieves 20.5%, representing a 5.5 percentage point improvement over the baseline while avoiding multi-sample decoding. This regime captures much of the structural benefit at lower cost and offers a practical accuracy–efficiency trade-off for deployment scenarios where budget constraints are critical. We report token usage directly, as in Appendix Table S2; exact dollar costs and wall-clock latencies were not consistently logged across providers, so Table 2 reports relative cost and latency expressions rather than provider-specific totals.

**External Structured Visual Reasoning.** To test whether the structured-hint effect is confined to ARC-AGI-2, we also evaluate an analogous structured pattern-detection pipeline on KiVA (Yiu et al., 2025), a visual analogy benchmark outside the ARC grid-puzzle setting. On 700 KiVA tasks, the pipeline improves All Correct over image-only baselines for Gemini 2.5 Flash (76.0% to 79.0%), Claude Sonnet 4.5 (30.7% to 51.0%), and GPT-4o (35.1% to 42.0%). Appendix Table S2 reports split accuracies and token usage. We present this as evidence of transfer to a second structured visual reasoning benchmark.

### 3.3 Implementation Details

The compositional reasoning pipeline is implemented using a heterogeneous ensemble of frontier LLMs together with deterministic grid-processing utilities. Grok-4 (`grok-4-0709`) serves as the primary solver and meta-classifier, responsible for generating output grids and selecting among candidate solutions. Pattern detection and structured hypothesis generation (Stage 2) are delegated to `o4-mini` via Azure OpenAI, using constrained structured outputs (`beta.chat.completions.parse`) to enforce JSON-formatted DSL detections. Low-level object extraction in Stage 1 (connected components, geometric statistics, and color counts) is computed algorithmically, while Claude Opus 4 (`claude-opus-4-20250514`) is used to enrich object descriptions and resolve ambiguous shape or cavity cues. Structured perception and validation calls use temperature 0, whereas solver generations used for self-consistency are sampled stochastically. Self-consistency decoding is applied in the final solving stage via $N$ independent samples with majority voting aggregated at the cell level; the number of solver attempts ranges from 3 to 10 per task, with at most 5 concurrent attempts per test case. Pattern detection is repeated 5 times, and only the top-3 patterns by detection count are forwarded to the solver, reducing noise. A rule-based jigsaw symmetry solver is triggered instead of the LLM when a symmetry score exceeds 0.70, providing a low-cost fallback. API reliability is maintained through key-pool rotation (up to 6 keys for Grok, 3 for Groq), capped exponential backoff with jitter (maximum 8 s), and a uniform request timeout of 72,000 s to accommodate long-horizon symbolic preprocessing. In the standalone public-evaluation run, 33 of the 37 solved tasks were solved through the DSL-guided pipeline, while 4 relied on LLM fallback generation from text hints. The appendix documents the implementation configuration, prompt templates, structured outputs, repetition protocol, self-consistency, and fallback behavior. Upon publication, we will release the parsing logic and deterministic object-extraction code; here, we separate deterministic grid utilities, heuristic fallbacks, and LLM-assisted analyses.

## 4 Related Work

**ARC and Systematic Generalization.** ARC was introduced to evaluate fluid intelligence under extreme data scarcity (Chollet, 2019). Each task requires inferring a latent transformation rule from a small number of demonstrations and applying it compositionally to unseen inputs. ARC-AGI-2 increases compositional depth and explicitly penalizes brute-force search and memorization strategies (ARC Prize Foundation, 2025b). ARC operationalizes systematic generalization—the ability to recombine learned structure in novel contexts—which remains a fundamental challenge for deep neural networks (Lake et al., 2017; Marcus, 2018; Battaglia et al., 2018). Although LLMs benefit from scaling laws (Kaplan et al., 2020) and chain-of-thought reasoning (Wei et al., 2022), performance on ARC-style abstraction remains far below human levels.

**Program Synthesis and DSL-Based Approaches.** Symbolic program synthesis methods search over domain-specific languages (DSLs) to construct programs consistent with demonstrations (Ellis et al., 2021; Devlin et al., 2017). These approaches provide interpretability and explicit compositional structure but suffer from combinatorial explosion as grid resolution and transformation depth increase. While effective on constrained tasks, exhaustive search becomes infeasible without strong structural priors.

**Neural and Relational Architectures.** End-to-end neural approaches attempt to learn direct grid-to-grid mappings using transformers or relational inductive biases (Vaswani et al., 2017; Santoro et al., 2017; Battaglia et al., 2018). Although such architectures demonstrate strong pattern recognition capabilities, they entangle perception and rule induction and often fail under compositional distribution shift. Object-centric and relational reasoning models emphasize explicit decomposition to improve systematic abstraction (Greff et al., 2020; Locatello et al., 2020), but typically lack strict cross-example constraint enforcement.

**LLM-Guided ARC Solvers.** Recent ARC solvers leverage large language models through prompting, iterative refinement, and self-consistency sampling (Wang et al., 2023; Zelikman et al., 2022). These systems improve empirical performance through extensive test-time sampling and probabilistic aggregation. However, they approximate consistency via stochastic voting rather than enforcing deterministic symbolic agreement.

**Neuro-Symbolic Reasoning.** Neuro-symbolic AI integrates neural perception with explicit symbolic reasoning to improve compositionality and interpretability (d'Avila Garcez et al., 2019; Kautz, 2022). Differentiable theorem provers (Rocktäschel & Riedel, 2017; Evans & Grefenstette, 2018), neural module networks (Andreas et al., 2016), and amortized program induction frameworks such as DreamCoder (Ellis et al., 2021) demonstrate that constraining hypothesis spaces with structured priors improves systematic generalization. These works collectively support architectures that separate perception from rule induction.

**Our Framework.** Our approach builds on these principles but differs by separating structured object abstraction, neural-guided hypothesis proposal over a fixed DSL, and cross-example consistency filtering. Rather than relying solely on unconstrained sampling, we use symbolic hints and consensus filtering to narrow the candidate set; the contribution is the measured comparison between this structured allocation of test-time compute and matched-budget frontier-model reasoning in the evaluated settings.

## 5 Limitations

Despite these advances, substantial limitations remain. Performance remains below human accuracy, and absolute scores have since been surpassed by subsequent systems, indicating that the current DSL is incomplete and that certain tasks require deeper relational abstraction. The reliance on self-consistency introduces a multiplicative inference cost, and the meta-classifier relies on candidate diversity rather than on program verification. Moreover, parts of the symbolic representation are LLM-assisted rather than fully formalized, so robustness still depends on the calibration of the prompting stack. We do not claim that neuro-symbolic reasoning, DSL-based program synthesis, or LLM-symbolic hybrids are new as broad categories. Qualitatively, failures arise from DSL expressivity gaps, ambiguous object or cavity extraction, unstable pattern detection across repeated calls, and final-solver failures to follow otherwise useful hints. ARC-AGI-2 and KiVA are public benchmarks, and some closed-source models used here postdate their release; we therefore cannot rule out benchmark contamination in the models' pretraining or post-training data, and do not claim a contamination-free evaluation. The phrase "no task-specific finetuning" refers only to our system: we do not finetune on evaluation tasks, but cannot certify the training data of the underlying closed-source models. Future work should therefore expand the expressivity of the transformation library, integrate refinement-based program search with symbolic intersection, develop more efficient mechanisms for enforcing consistency without heavy sampling, and replicate the study with models known to predate the benchmarks.

## 6 Conclusion

ARC-AGI-2 exposes a central tension in contemporary AI systems: scale alone does not yield systematic abstraction. Within the evaluated ARC-AGI-2 and KiVA settings, our results support a structural insight rather than only a numerical one: explicit separation of perception, hypothesis generation, and symbolic constraint can improve performance relative to unstructured test-time scaling. By grounding reasoning in object-level structure and restricting transformations to a compositional DSL, the system reduces hypothesis entropy and encourages cross-example invariance. The observed gains are therefore consistent with the value of inductive bias rather than brute-force search or scaling context and samples alone. This suggests that structured visual reasoning systems may benefit from architectural priors that explicitly encode compositional structure, rather than relying only on larger models or longer contexts.

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

Table S1: Implementation configuration for the compositional neuro-symbolic pipeline. Roles are assigned per stage; deterministic grid utilities are used wherever possible, and LLMs supply structured analyses.

| Category | Parameter | Value / Description |
|---|---|---|
| *Model Assignment* | | |
| | Primary solver & meta-classifier | Grok-4 (`grok-4-0709`), xAI API |
| | Pattern detection (Stage 2) | `o4-mini-2025-04-16`, Azure OpenAI; structured outputs via `beta.chat.completions.parse` |
| | Low-level object extraction (Stage 1) | Deterministic grid processing for connected components, bounding boxes, centroids, and color histograms |
| | Object metadata enrichment (Stage 1) | Claude Opus 4 (`claude-opus-4-20250514`), native Anthropic client for ambiguous shape/cavity cues |
| | Pattern detection fallback | `openai/gpt-oss-120b` via Groq and Together AI |
| *Inference Hyperparameters* | | |
| | Temperature | 0.0 for structured perception/validation calls; non-zero stochastic sampling for solver self-consistency runs |
| | Max tokens — Claude Opus 4 (Stage 1) | 32,000 |
| | Max tokens — `o4-mini` (Stage 2) | 4,096 (object analysis); 100,000 (experiments) |
| | Max tokens — Groq / Together AI | 20,000 (summarization); 100,000 (Together AI) |
| | Max tokens — end-to-end solver | 2,000 |
| | Anthropic thinking budget | 16,000 tokens |
| | Groq / Together `reasoning_effort` | `"high"` |
| *Concurrency & Retry* | | |
| | Pattern detection concurrency | 5 parallel calls (env: `OPENAI_CONCURRENCY`) |
| | Summary concurrency | 60 parallel calls (env: `OPENAI_SUMMARY_CONCURRENCY`) |
| | Pattern detection repetitions | 5 per example (env: `PATTERN_DETECTION_REPETITIONS`) |
| | Solver attempts per task | 3–10 (passed as argument) |
| | Max concurrent solver attempts | 5 per test case (hardcoded) |
| | API max retries | 3 (env: `OPENAI_MAX_RETRIES`) |
| | Backoff on retryable errors | Capped at 8.0 s with jitter |
| | Backoff on general errors | 0.1 s (fast retry) |
| | Request timeout | 72,000 s (env: `OPENAI_TIMEOUT_SECONDS`) |
| *API Key Management* | | |
| | Grok key pool | Up to 6 keys; random selection per async call (`XAI_API_KEYS`) |
| | Groq key pool | 3 keys; random selection per call (`GROQ_API_KEYS`) |
| | Request jitter | 0.05–0.1 s (Groq/Together), 0.2–0.8 s (OpenAI) |
| *Decoding & Aggregation* | | |
| | Self-consistency voting | Majority vote at cell level across $N$ grid predictions |
| | Top-$k$ pattern filtering | Top-3 patterns by detection count forwarded to solver |
| | Jigsaw symmetry threshold | Score $> 0.70$ triggers rule-based solver (bypasses LLM) |

Table S2: KiVA validation on 700 tasks. All Correct requires all three MCQ stages to be correct. Max tokens per call is 32,000 with three attempts per stage. Baselines use images only without hints.

| Model | Setting | Cross-Domain (%) | Within-Domain (%) | Extrap. (%) | All Correct (%) | Avg. Tokens |
|---|---|---|---|---|---|---|
| Gemini 2.5 Flash | Image-only baseline | 99.3 | 81.9 | 87.0 | 76.0 | 7,844.3 |
| Gemini 2.5 Flash | Structured pipeline | 99.0 | 84.0 | 90.0 | 79.0 | 10,218.6 |
| Claude Sonnet 4.5 | Image-only baseline | 99.3 | 81.0 | 39.3 | 30.7 | 4,716.5 |
| Claude Sonnet 4.5 | Structured pipeline | 99.0 | 82.0 | 57.0 | 51.0 | 7,429.9 |
| GPT-4o | Image-only baseline | 96.7 | 65.1 | 54.9 | 35.1 | 7,776.6 |
| GPT-4o | Structured pipeline | 96.0 | 66.0 | 63.0 | 42.0 | 10,861.6 |

# A  Atomic Transformation Library (Unit Patterns)

Our structured reasoning pipeline is built upon a curated library of 22 atomic transformations, or "Unit Patterns," that serve as the Domain-Specific Language (DSL) for ARC. These patterns (visualized in Figure 2) were identified through manual analysis of the ARC-AGI-1 and ARC-AGI-2 training sets. For DSL pattern design and tuning, we used 100 examples from the training set; the test set was not used. They are designed as a compact "core knowledge" set covering many recurring primitive and compositional operations seen in the corpus, while remaining incomplete.

Unlike simple primitives (e.g., `Rotate90`), these Unit Patterns are more complex, parameterized operations that describe a common reasoning process found in ARC tasks. They are the target operations our `o4-mini` hypothesis generator (Stage 2) attempts to detect.

The 22 patterns are as follows:

- **`Horizontal Fill`**
    - **Description:** Extend or fill an object horizontally across contiguous empty or target cells.
    - **Parameters:**
        * source_object: ["line", "square", "rectangle", "cavity"]
        * column_index: ["left of an object", "right of an object"]
        * fill_color: ["based on source", "based on some different objects"]
        * sequence: ["based on source width", "based on source height"]
        * stop_condition: ["another object", "grid boundary", "specific color"]
        * overlaps: ["keep the latest", "no overlaps possible"]

- **`Vertical Fill`**
    - **Description:** Extend or fill an object vertically across contiguous empty or target cells.
    - **Parameters:**
        * source_object: ["line", "square", "rectangle", "cavity"]
        * row_index: ["top of an object", "below an object"]
        * fill_color: ["based on source color"]
        * sequence: ["based on source width", "based on source height"]
        * stop_condition: ["another object", "grid boundary", "specific color"]

- **`Connecting Bridges`**
    - **Description:** Draw a "bridge" (line/shape) between two objects in a specified color order.
    - **Parameters:**
        * source_object: ["line", "square", "rectangle", "cavity"]
        * target_object: ["line", "square", "rectangle", "cavity"]
        * bridge_color: ["based on bridge starting point", "based on bridge ending point", "based on cavity inside an object"]
        * connection_shape: ["line", "triangle", "rectangle", "circle"]
        * path_direction: ["orthogonal", "diagonal", "based on color sequence"]
        * thickness: ["based on width of cavity", "based on width of starting object"]

- **`Boundary Attachment Fill`**
    - **Description:** Close holes or voids inside an object's boundary bounding area.
    - **Parameters:**
        * objects_with_holes: ["horizontally laid", "vertically laid", "diagonally laid"]
        * attachment_direction: ["left", "right", "top", "bottom"]
        * fill_logic: ["fits in space to form rectangle", "gets laid on the object"]

         * object_filled: ["irregular", "triangle", "rectangle", "square"]

- **Diagonal Fill**

  - **Description:** Propagate color or object along a diagonal axis.
  - **Parameters:**
    * source_point_or_corner: ["L-shaped", "rectangle"]
    * direction: ["bottom-right", "top-left", "top-right", "bottom-left"]
    * fill_color: ["same as source", "complementary to source", "change on bounce"]
    * stop_condition: ["object obstruction", "hit grid boundary"]

- **Pattern Matching Fill / Remove**

  - **Description:** Identify a repeating subpattern and either color it in or erase it.
  - **Parameters:**
    * template_pattern: ["alternate objects", "similar objects", "symmetry via some axis"]
    * operation: ["remove cells to match pattern", "fill cells to match pattern"]
    * fill_color: ["boundary color", "pattern color"]
    * tolerance: ["no tolerance", "edges are exceptions"]
    * target_regions: ["inside a cavity", "outside an object"]

- **Creating Patterns based on starting Objects**

  - **Description:** Generate a larger or repeated pattern seeded from one or more "starter" objects.
  - **Parameters:**
    * seed_objects: ["colored cell", "rectangle", "diagonal"]
    * transformation_sequence: ["circular", "straight", "fill all", "towards an object"]
    * inter_object_spacing: ["none", "single", "multiple", "variable"]
    * repeat: ["till filling the cavity", "only once"]
    * stopping_condition: ["reached an object", "reached boundary", "filled object completely"]

- **Find Objects in the Input Image and Color Them**

  - **Description:** Detects all instances of a certain object class and applies a new color.
  - **Parameters:**
    * object_type: ["plus", "rectangle", "irregular", "circle", "cell", "horizontal bar"]
    * new_color: ["complements the original color", "constant throughout", "alternating pattern"]
    * detection_method: ["exact match", "fuzzy", "at some location"]
    * overlap_policy: ["all unique", "overlaps allowed"]

- **Remove Objects from the Output in a Particular Sequence**

  - **Description:** Systematically delete objects one at a time in a defined order.
  - **Parameters:**
    * object_list_ordered: ["all in the row", "all in a column", "same shape"]
    * removal_method: ["erase and color", "replace with background"]
    * trigger_condition: ["based on an object", "leftmost", "rightmost", "topmost", "overlaps"]

- **Rearrange the Objects in the Output in a Particular Sequence/Pattern**

  - **Description:** From a set of objects, only retain those in a given order, rearrange the rest.
  - **Parameters:**

- * keep_sequence: ["ascending order of height", "descending order of height"]
  * color_of_object: ["same as in-place object", "original color"]
  * pattern: ["to a particular part of another object", "to a particular region"]

- **Alternating Pattern Filling**

  – **Description:** Fill cells with two (or more) colors/objects in an alternating rhythm (checkerboard, stripes).
  – **Parameters:**
    * colors: [["A", "B"], ["A", "A", "B"]]
    * pattern_type: ["checkerboard", "stripe_vertical", "stripe_horizontal"]
    * internal_sequence_spacing: ["none", "singular"]

- **Object Translation Based on Environment Colors**

  – **Description:** Move an object to a place based on the colors surrounding them.
  – **Parameters:**
    * moving_object_shape: ["plus", "square", "rectangle", "all cells"]
    * target_environment_color: ["same as moving object", "complementary color"]
    * translation_vector: ["centroid of the environment colors", "on top of environment color"]
    * step_size: ["arbitrary", "fixed size"]

- **Cavity Fill**

  – **Description:** Fill the cavities inside bigger objects.
  – **Parameters:**
    * object_outline: ["U shaped", "V shaped", "rectangular", "triangle", "square"]
    * max_indent_depth: ["based on available filling material", "till complete object"]
    * fill_color: ["arbitrary", "based on material already present"]

- **Add/Replace an Object**

  – **Description:** Swap out one object for another, preserving position or properties.
  – **Parameters:**
    * source_object: ["horizontal bar", "vertical bar", "rectangle", "square", "circle", "triangle", "irregular"]
    * add_replacement_object: ["horizontal bar", "vertical bar", "rectangle", "square", "circle", "triangle", "cell"]
    * inherit_properties: ["same midpoint", "same centroid", "at some location"]
    * additional_change: ["add a boundary to new object", "do nothing"]

- **Falling Down (Gravity-Effect)**

  – **Description:** Let objects "drop" vertically until they hit another object or the floor.
  – **Parameters:**
    * object_list: ["cell", "square", "rectangle"]
    * gravity_direction: ["downward"]
    * collision_map: ["horizontal bar", "vertical bar"]

- **Get Attached to Similar Object**

  – **Description:** Move or grow an object until it contacts another of the same type.
  – **Parameters:**
    * moving_object: ["plus", "U shaped", "V shaped", "square", "rectangle", "irregular"]
    * target_object_type: ["rectangle", "square", "irregular"]

      ∗ attachment_rule: ["head on with common color side", "fit into cavity"]

      ∗ movement_path: ["fixed numeric steps", "reach goal"]

- Object Translation Based on Goal

  - **Description:** Move objects toward a specified "goal" region or object.
  - **Parameters:**
    - ∗ source_object: ["square", "rectangle", "irregular"]
    - ∗ goal_location_or_object: ["square", "matching pattern"]
    - ∗ pathfinding_method: ["straight-line", "fixed path"]
    - ∗ step_count_or_speed: ["stop on obstacle", "stop on goal", "fixed"]

- Object Dismantles

  - **Description:** Break an object into constituent parts or pixels.
  - **Parameters:**
    - ∗ source_object: ["irregular", "rectangular", "square"]
    - ∗ fragment_shape: ["individual cells", "smaller tiles", "break at hit"]
    - ∗ dismantle_sequence: ["outer-to-inner", "when hit by other object", "symmetric"]
    - ∗ dispersion_pattern: ["momentum conserved", "toward hit object", "away from hit object"]

- Symmetry-Based Pattern

  - **Description:** Reflect or rotate objects/patterns around an axis or point.
  - **Parameters:**
    - ∗ symmetry_type: ["horizontal", "vertical", "rotational"]
    - ∗ axis_or_center_point: ["horizontal bar", "vertical bar", "single cell"]
    - ∗ object_group: ["individual cells", "square"]
    - ∗ copy_mode: ["duplicate", "mirror"]

- Ray-Cast / Ray-Trace Pattern

  - **Description:** Project a "ray" from a source until it hits a wall or object, marking its path in a shape.
  - **Parameters:**
    - ∗ ray_source: ["starting cell", "object"]
    - ∗ direction: ["horizontal", "vertical", "diagonal", "change on hit"]
    - ∗ shape: ["line", "triangle", "circle", "rectangle"]
    - ∗ stop_condition: ["object", "boundary"]
    - ∗ mark_color: ["same as starting point", "alternating pattern", "change on hit", "based on other objects"]

- Scattering Pattern

  - **Description:** Project a scatter-like pattern which is triangular in shape with staircase-like edges, and fills all the cells in its path.
  - **Parameters:**
    - ∗ source: ["starting cell", "object"]
    - ∗ direction: ["horizontal", "vertical", "diagonal", "radially outwards"]
    - ∗ shape: ["triangle"]
    - ∗ stop_condition: ["object", "boundary"]
    - ∗ mark_color: ["same as starting point", "alternating pattern", "change on hit", "based on other objects"]
    - ∗ boundary: ["single cell thickness of different color than the pattern", "multi cell thickness of different color than the pattern"]

* edge_pattern: ["staircase with a width 'w' and height 'h', where 'w' and 'h' are number of cells"]

- **Patterns formed using small objects**
  - **Description:** Spatial patterns and color scheme formed by smaller objects.
  - **Parameters:**
    * small_object_type: ["small adjacent objects", "parts of a bigger object"]
    * small_pattern_type: ["spatial pattern and/or color scheme pattern formed by smaller distinct objects", "coloring scheme pattern formed inside a object"]

# B Prompt Templates

## B.1 Solver Prompt Template

This prompt is used to guide the model in performing step-by-step reasoning and transformation inference based on provided Input–Output grid examples and structured hints.

```
Example {i}:
Input:
{input_viz}

Output:
{output_viz}
"""

solver_prompt_template = """Consider the following examples:

{examples}

Based on the pattern in the examples, what would the output for the following
test input be?

Test input:
{test_input_viz}

Test output:

(Hint: {hint})
"""

new_solver_prompt = """
You are given a set of Input-Output Examples and a detailed list of
Transformation Steps (the Hint).

## Your Task

1. Examine the examples to understand exactly how the transformations
are applied.
2. Follow the hint steps in order, exactly as described, with
no additions or omissions.
3. Use the examples to resolve any remaining ambiguities in the hint.
4. Apply the same sequence of transformations to the Test Input.

> Note: The Test question is slightly more challenging than the training
examples-it may involve rotation invariance and color invariance.

---

## Inputs Provided

 Examples: '{examples}'
 Hint (Transformation Steps): '{hint}'
 Test Input Visualization: '{test_input_viz}'
---
```

## Output Instructions

1. Present your full reasoning, detailing how each step from the hint maps
to the transformation operations you perform.
2. Do not invent any new rules or skip any hint steps. Ensure the Test Output
reflects the same behavior demonstrated by the examples.
3. Embed the Final output in ``` ``` and use \n for new row and | as column
separator.
4. I want you to solve this puzzle step by step. First, restate the problem.
Then outline your plan. Then execute each step, numbering them, and finally give
your answer.
Test Output:

### B.2 Pattern Detection Prompt

This prompt is designed for object-level pattern recognition, enabling the model to analyze Input and Output
grids and return structured JSON detections for each candidate pattern.

Coordinate System:

 Top-left cell is (0, 0). x increases down (rows), y increases right (columns).

Given:

 Input Grid:
`{}`
 Output Grid:
`{}`
 Input Objects: `{}`
 Output Objects: `{}`
 Pattern Specifications: `{}`

Task: For each pattern in the given list:

1. Compare Input vs Output objects to identify moves, removals, additions,
rotations, shifts, duplications, or color changes.
2. Do note that some objects might combine to form a multi color bigger object.
3. Decide if the pattern applies.
4. Provide a concise reason for your decision even if `pattern_detected`
is `false`. Include the precise reasons for object movements, additions, removals,
retention. There exist some logic, your task is to find it using the help of patterns
and params.
5. List only the matched parameter values under `params` (use an empty object if none).

Output: Return only this JSON array (no extra text):

```json
[
  {
    "reason": "<detailed explanation>",
    "pattern_detected": <true|false>,
    "pattern_name": "<Pattern Name>",
```

```
  "pattern_description": "<Pattern Specification.description>",
  "params": { / matched values or {} / }
 },
 ...
]
```

## B.3 Meta-Classifier Prompt

This prompt is used by the final ensemble model (Section 2). It instructs Grok-4 to analyze the training task, the test task, and a list of 4 candidate solutions (2 from the Compositional Reasoner, 2 from the ARC Lang Solver) and select the single most likely solution. The *pass@2* submission is obtained by running this classifier twice, removing the first selected candidate before the second pass.

```
"""You are given a some input-output examples, and a test task. You are also
given a list of possible solutions.
Your job is to figure out the pattern in the input-output examples
and select the most likely solution from the list
of possible solutions. Output only the solution ID.

Input-output examples:
{train_tasks_prompt}

Test task:
{test_task_viz}

Possible solutions:
{answers_viz}

Enter your answer in this format:
***Solution ID***
"""
```

## B.4 Object Detection Prompts

These prompts augment deterministic grid-derived features in the initial object abstraction stage (Stage 1). They are called by the 'Grid' and 'BaseObject' classes defined in our system to provide fine-grained feature analysis when heuristic extraction is ambiguous or when richer textual descriptors are useful downstream.

### B.4.1 Background Color Detection Prompt

This prompt is used by the 'Grid.find-background' method as a fallback to identify the most likely background color when simple frequency/edge heuristics are ambiguous.

```
You are given a 2D grid of integers ranging from 0 to 9.
Each integer represents a color. The goal is to determine the most likely
background color in the grid.

The background color is typically:
- The color that appears most frequently overall in the grid.
- The color that touches the edges of the grid (top, bottom, left, or right).
- The color that is not part of compact or enclosed clusters (i.e., likely
  not part of foreground objects).

Use the following decision strategy:
1. Start by identifying the most frequent color in the grid.
```

```
2. If that color also touches one or more edges of the grid, assume it is
   the background.
3. If multiple colors meet these criteria or the result is ambiguous, return -1.

Only return a JSON object in this exact format:
{"background_color": <integer from 0 to 9 or -1>}

Do not explain your reasoning before the JSON. Only output the JSON on
the first line.

Here is the grid:
{grid_str}
```

### B.4.2 Object Shape Analysis Prompt

This prompt is used by the 'BaseObject.-analyze-shape' method to generate a natural language description of an object's shape, which can be used in downstream reasoning.

```
You are given a rectangular 2D grid of shape {grid_shape}.
Each pixel is represented by an integer between 0 and 9, where 0 means black
(background), and other values are colors. The pixels in colors represent
an object
Here is the full grid:
{grid_str}

The object is defined by the following coordinates:
{coord_str}

Your task is to analyse the shape of this object, remember that the object
can also be irregular and have cavities inside it.
Describe the shape of this object in a single statement, try to include as
much detail as possible
```

### B.4.3 Cavity Detection Prompt

This prompt is used by the 'BaseObject.get-cavities' method. It uses the shape analysis (if available) to identify and extract the coordinates of any enclosed cavities (holes) within an object.

```
You are given a rectangular 2D grid of shape {grid_shape}.
Each pixel is represented by an integer between 0 and 9, where 0 means black
(background), and other values are colors. The pixels in colors represent
an object
Here is the full grid:
{grid_str}

The object is defined by the following coordinates:
{coord_str}

Here is the analysis of the shape of the object: {shape_analysis}
Your task is to find out the cavities inside this object and return the
coordinates that constitute the object.
Return the output in this format:
Cavity 1 : <List of coordinates of Cavity 1>
Cavity 2 : <List of coordinates of Cavity 2 >
....................
```

```
Cavity n : <List of coordinates of Cavity n>
where 'n' is the number of cavities you detected. Return the coordinates
only, don't give any explanation.
Use only plain ASCII characters, standard spaces, parentheses, and commas
exactly as shown.
Do not include any special Unicode spaces or extra formatting.
```

