# OpenReview forum: "Compositional Neuro-Symbolic Reasoning"
_TMLR — Under review for TMLR_

### Review · Reviewer_gQga · 2026-05-25

**Summary Of Contributions:**

This paper presents an ARC-AGI-2 solving pipeline that extracts object-level grid information, uses LLMs to identify transformations from a manually defined pattern library, and provides these patterns as hints to a final LLM solver. It reports 24.4% pass@2 accuracy for the standalone pipeline and 30.8% when combined with ARC Lang Solver through a meta-classifier. The system is reasonably documented, but its contribution appears to be benchmark-specific structured prompting rather than the executable neuro-symbolic reasoning framework suggested by the presentation.

**Audience:**

No

**Audience Explanation:**

Although ARC-AGI-2 is relevant to some researchers, the paper provides limited insight beyond task-specific engineering on an artificial grid-puzzle benchmark. The connection between improvement on this task and broader real-world reasoning capability is unclear, and the work does not offer a theoretical contribution or a sufficiently general validated method. I therefore do not find the current findings interesting for TMLR audience.

**Claims And Evidence:**

No

**Claims Explanation:**

The main claims are not adequately supported. The paper formally describes executable DSL programs, symbolic verification, and cross-example consistency filtering, but the implemented system primarily uses repeated LLM pattern detections as hints for another LLM solver. The evaluation does not isolate the effects of the handcrafted pattern library, multiple model calls, self-consistency, rule-based fallback, or ensemble selection. In addition, results are reported only on the public ARC-AGI-2 evaluation set. The evidence supports a narrower claim that structured hints improve this particular solver on this benchmark, but not broader claims about neuro-symbolic reasoning or systematic generalization.

**Requested Changes:**

The main concern is fundamental rather than a matter of presentation or limited ablation. The paper would need to demonstrate scientific insight beyond benchmark-specific improvements on ARC-style grid puzzles, through broader validation or a substantive methodological or theoretical contribution. Without this, the current results are insufficient to support acceptance.

---

> ### Author Response · Authors · 2026-06-26
> **Rebuttal to Reviewer gQga**
>
> We thank Reviewer gQga for the precise articulation of the central concerns.
>
> **Formalism-vs-implementation gap.**
>
> The original methodology described executable DSL programs, symbolic verification, and cross-example intersection in overly deterministic terms. This could give the impression that every step is strict symbolic execution, whereas the implemented system is hybrid: deterministic grid processing and executable checks are combined with LLM-assisted pattern detection, structured outputs, consensus filtering, and DSL-guided final generation. The revised paper will distinguish these cases. For deterministic primitives, consistency can be checked by execution. For LLM-assisted primitives, the model returns structured pattern hypotheses and parameters, which are retained only when they satisfy explicit contracts such as schema validity, object/color/shape consistency, and agreement with the observed examples, where applicable. Stage 4 is described as a DSL-guided solution synthesis rather than guaranteed closed-form symbolic rendering. We will also report the run-log breakdown: among the 37 solved ARC-AGI-2 tasks, 33 were solved through the DSL-guided pipeline, and 4 relied on LLM fallback generation from text hints.
>
> **Contributions of the system components.**
>
> The reviewer is right that the original evaluation did not sufficiently separate the handcrafted pattern library, multiple model calls, self-consistency, rule-based fallback, and ensemble selection. The revised paper will separate these claims more carefully. The standalone Compositional Reasoner result remains 24.4%, while the 30.8% result is explicitly identified as the ARC Lang Solver meta-classifier ensemble. The ensemble is therefore not used as evidence for the standalone method. We also add a matched-token-budget comparison against the strongest frontier-model baselines available before September 2025. Under this protocol, our standalone structured DSL-guided pipeline obtains 24.4%, compared with GPT-5 at 18.3%, Grok 4 at 16.0%, and Claude Opus 4 at 8.6%. Under this evaluation setup, explicit structure and DSL-guided reasoning outperform the matched-budget frontier-model baselines. We do not claim that this eliminates every possible confound or that every component contributes independently.
>
> **Benchmark-specificity and broader validation.**
>
> The reviewer’s concern is that experiments were limited to ARC-AGI-2. To address this, we add an evaluation on KiVA, a structured visual analogy benchmark outside the ARC-AGI-2 grid-puzzle setting. We use an analogous structured pattern-detection pipeline with KiVA-specific unit patterns.
>
> **Table 1:** KiVA validation (700 tasks; All Correct requires passing all 3 MCQ stages; max tokens/call 32,000; 3 attempts per stage):
>
> | Model | Setting | Cross-Domain | Within-Domain | Extrapolation | All Correct | Avg Tokens/Task |
> |---|---|---:|---:|---:|---:|---:|
> | Gemini 2.5 Flash | Image-only baseline | 99.3% | 81.9% | 87.0% | 76.0% | 7,844.3 |
> | Gemini 2.5 Flash | Neuro-symbolic pipeline | 99.0% | 84.0% | 90.0% | 79.0% | 10,218.6 |
> | Claude Sonnet 4.5 | Image-only baseline | 99.3% | 81.0% | 39.3% | 30.7% | 4,716.5 |
> | Claude Sonnet 4.5 | Neuro-symbolic pipeline | 99.0% | 82.0% | 57.0% | 51.0% | 7,429.9 |
> | GPT-4o | Image-only baseline | 96.7% | 65.1% | 54.9% | 35.1% | 7,776.6 |
> | GPT-4o | Neuro-symbolic pipeline | 96.0% | 66.0% | 63.0% | 42.0% | 10,861.6 |
>
> These results show that the structured-hint effect is not confined to ARC-AGI-2. KiVA establishes transfer to a second structured visual reasoning benchmark, not broad real-world generality. We also report token usage to make the additional computation visible.
>
> **Paper’s scientific contribution.**
>
> We have reframed the paper away from the broad claim that “neuro-symbolic reasoning solves ARC.” The revised contribution is narrower: structured compositional inductive bias, implemented through object abstraction, DSL pattern detection, and cross-example consistency filtering, improves LLM-based visual reasoning relative to matched stochastic/chain-of-thought scaling in our evaluated settings. This is the scientific insight we intend the paper to support, and not the claim that neuro-symbolic reasoning or LLM-symbolic hybrids are new as categories.
>
> **Summary**
>
> We understand the reviewer’s concern that an ARC-only engineering result may not be of broad TMLR interest. The revised framing and KiVA validation are intended to address this directly. The paper is now positioned as evidence about the value of structural inductive bias relative to test-time stochastic scaling, evaluated on two structured visual reasoning benchmarks, rather than as a benchmark-specific ARC solver alone.

---

### Review · Reviewer_iBbk · 2026-05-28

**Summary Of Contributions:**

This work proposes a neuro-symbolic framework to solve the ARC-AGI-2 benchmark. The key idea is to obtain abstract low-dimensional information from the inputs so that LLMs can generate symbolic programs to solve the task. There are two major weaknesses: (1) the idea of neuro-symbolic reasoning is not new (some works are mentioned in the related works section), and (2) the approach is tailored to the ARC-AGI-2 benchmark. There is no external applicability in its current form.

**Audience:**

Yes

**Audience Explanation:**

If the claim can be made clearly, I think this work will be of interest to at least some members of the TMLR audience.

**Claims And Evidence:**

No

**Claims Explanation:**

The claim is that neuro-symbolic reasoning with LLMs' support can solve the ARC-AGI-2 benchmark. Neuro-symbolic reasoning itself is not new, and there is no contribution in that angle. Symbolic reasoning has been used to solve the ARC-AGI-1 benchmark (see icecuber's solution that scored the highest in the ARC-AGI-1 Kaggle competition). I am not sure whether reusing ideas to solve a particular benchmark constitutes a valid claim as per TMLR guidelines.

That said, I am not aware of any works that combine LLMs with symbolic logic (not my domain). So, I will be willing to change my decision on this if the authors can (1) confirm the novelty of LLM+symbolic, and (2) show that their proposed approach can work on benchmarks other than those in the ARC framework.

**Requested Changes:**

See my comment on whether the claim is supported or not. Please let me know if my understanding of the paper is incomplete.

1. Is the proposed benchmark supposed to be of general use, or only for ARC-AGI-2? If so, can you show it working on a few more benchmarks, preferably real-world ones?

2. Is there any novelty in the proposed approach compared to the existing neuro-symbolic literature? Have LLMs been used with symbolic AI before?

3. The presentation in the introduction section can be improved for the readers who are not familiar with the exact input-output formulations of ARC-AGI. Maybe a few samples from the ARC-AGI benchmark, along with how existing models answer them, can be shown in the introduction.

---

> ### Author Response · Authors · 2026-06-26
> **Rebuttal to Reviewer iBbk**
>
> We thank Reviewer iBbk for the clear and actionable review. We address the main concerns directly: novelty relative to prior neuro-symbolic/ARC systems, and applicability beyond ARC-AGI-2.
>
> **Is the proposed benchmark supposed to be of general use, or only for ARC-AGI-2? If so, can you show it working on a few more benchmarks, preferably real-world ones?**
>
> We have added an evaluation on KiVA, a structured visual analogy benchmark outside the ARC-AGI-2 grid-puzzle setting. We use an analogous structured pattern-detection pipeline with KiVA-specific unit patterns and compare it against image-only baselines on 700 tasks. The improvement is consistent across three model families:
>
> **Table 1:** KiVA validation (700 tasks; All Correct requires passing all 3 MCQ stages; max tokens/call 32,000; 3 attempts per stage):
>
> | Model | Setting | Cross-Domain | Within-Domain | Extrapolation | All Correct | Avg Tokens/Task |
> |---|---|---:|---:|---:|---:|---:|
> | Gemini 2.5 Flash | Image-only baseline | 99.3% | 81.9% | 87.0% | 76.0% | 7,844.3 |
> | Gemini 2.5 Flash | Neuro-symbolic pipeline | 99.0% | 84.0% | 90.0% | 79.0% | 10,218.6 |
> | Claude Sonnet 4.5 | Image-only baseline | 99.3% | 81.0% | 39.3% | 30.7% | 4,716.5 |
> | Claude Sonnet 4.5 | Neuro-symbolic pipeline | 99.0% | 82.0% | 57.0% | 51.0% | 7,429.9 |
> | GPT-4o | Image-only baseline | 96.7% | 65.1% | 54.9% | 35.1% | 7,776.6 |
> | GPT-4o | Neuro-symbolic pipeline | 96.0% | 66.0% | 63.0% | 42.0% | 10,861.6 |
>
> 'All Correct' means the model must pass all three MCQ stages. We also report average token usage in the revised table to make the additional computation transparent. KiVA shows that the structured-hint effect is not limited to ARC-AGI-2, but we do not claim this establishes broad general-purpose or real-world applicability. Because ARC-AGI-2 and KiVA are public benchmarks, and some closed-source models may have been trained or post-trained after these data were publicly available, we also do not claim a contamination-free evaluation. We will also clarify in the revised manuscript that the paper does not propose a new benchmark; it proposes a method evaluated on existing benchmarks.
>
> **Is there any novelty in the proposed approach compared to the existing neuro-symbolic literature? Have LLMs been used with symbolic AI before?**
>
> The reviewer’s understanding is partly correct: we do not claim that neuro-symbolic reasoning, DSL-based solving, or the combination of LLMs with symbolic methods is new as a broad category. Prior work has combined LLMs with program synthesis, verifiers, symbolic search, and DSL-based solvers; symbolic systems have shown the strength of hand-engineered symbolic search on ARC-AGI-1. We will revise the related-work section to make this positioning explicit rather than implying that the general paradigm is novel. The contribution we intend to claim is empirical. The paper studies whether adding a structured compositional inductive bias, implemented through object abstraction, DSL pattern detection, and cross-example consistency filtering, improves LLM reasoning beyond stochastic test-time sampling on the same task family. In the revised manuscript, we will frame the result as a matched-token comparison: our standalone structured DSL-guided pipeline obtains 24.4%, compared with the strongest pre-September 2025 frontier-model baselines under the same token budget: GPT-5 at 18.3%, Grok 4 at 16.0%, and Claude Opus 4 at 8.6%. Thus, the contribution is not ‘neuro-symbolic reasoning is new,’ but rather evidence that explicit structure and DSL-guided reasoning can outperform matched-budget frontier-model baselines under this evaluation setup.
>
> **The presentation in the introduction section can be improved for the readers who are not familiar with the exact input-output formulations of ARC-AGI. Maybe a few samples from the ARC-AGI benchmark, along with how existing models answer them, can be shown in the introduction.**
>
> We agree that the introduction assumed too much ARC familiarity. The revised introduction will explain the ARC-AGI input-output format more concretely and include illustrative examples showing the training grids, the latent transformation to infer, and how the test grid is solved. This is intended to make the task formulation clear before introducing the pipeline.

---

> > ### Comment · Reviewer_iBbk · 2026-06-28
> > **Reviewer's response to authors' rebuttal**
> >
> > I thank the authors for their response.
> >
> > I couldn't find the new results or the changes in the PDF. Is it not updated?
> >
> > Also, while it is good to see new results on a new benchmark, I doubt if it is a valuable contribution to combine existing neuro-symbolic ideas to solve contemporary benchmarks. The benchmarks may, and will likely, change, and the leading benchmarks to track AGI progress in the future may not even take the form of an abstract reasoning benchmark.

---

> > > ### Author Response · Authors · 2026-06-29
> > > **Response to Reviewer's Concern**
> > >
> > > Thank you for the follow-up. You are correct that the currently visible PDF has not yet been updated. The new KiVA results and the manuscript changes were described in the rebuttal, and they will be incorporated in the revised/camera-ready version.
> > >
> > > On the broader point, we agree with the premise that benchmarks change and that future benchmarks for tracking AGI progress may look very different from ARC-style abstract reasoning tasks. Our intended claim is narrower and more portable: in our evaluated settings, allocating test-time compute to structured abstraction and constrained hypothesis generation improves over allocating it only to unstructured stochastic sampling/frontier-model reasoning.
> > >
> > > The role of ARC-AGI-2 and KiVA is evidential rather than definitional. ARC-AGI-2 is an abstract program-synthesis-lineage benchmark, while KiVA is a distinct visual analogy benchmark with everyday-object stimuli and a different evaluation structure. We use them to test whether the structure-versus-sampling effect is confined to one benchmark family. The KiVA result does not establish broad general reasoning ability, but it does show that the effect is not limited to ARC-AGI-2 grid puzzles.
> > >
> > > On whether combining existing neuro-symbolic components constitutes a contribution: we agree that neuro-symbolic reasoning itself is not new, and we do not claim novelty at that level. The contribution we aim to support is the measured comparison: under a matched token budget, our explicit structure/DSL-guided pipeline improves over matched-budget stochastic frontier-model baselines in these visual reasoning settings. We think this is of interest because it addresses an active question in current ML systems: whether test-time compute is better spent on more sampling or on structured abstraction and constrained hypothesis generation.
> > >
> > > We appreciate the exchange. The specific DSLs may not transfer unchanged to future benchmarks, and we will state that limitation clearly; the broader design principle we test is the value of structured inductive bias relative to unstructured test-time scaling.

---

> > > > ### Comment · Reviewer_iBbk · 2026-06-30
> > > > **Reviewer's response**
> > > >
> > > > I again thank the authors for the effort they have put into the work and the rebuttal.
> > > >
> > > > TMLR allows PDFs to be edited during the rebuttal. I believe this policy is to allow the reviewers to see the exact changes made by the authors in response to the rebuttal. So I would appreciate it if the authors could make the above-mentioned changes in the PDF.
> > > >
> > > > About scope and contribution:
> > > >
> > > > > Our intended claim is narrower and more portable: in our evaluated settings, allocating test-time compute to structured abstraction and constrained hypothesis generation improves over allocating it only to unstructured stochastic sampling/frontier-model reasoning.
> > > >
> > > > This is not a new claim, as this is exactly what other neuro-symbolic reasoning papers implicitly support. This exact claim could be repeated for a new task tomorrow without any change in the method.

---

> > > > > ### Author Response · Authors · 2026-07-02
> > > > > **Response to Reviewer**
> > > > >
> > > > > Thank you again. We have now uploaded the revised PDF, so the changes are visible.
> > > > >
> > > > > We agree that the general principle “structure helps reasoning” is not new. What we think is useful here is the contemporary empirical test of that principle against frontier LLM test-time scaling: under a matched inference budget, the structured pipeline improves over strong stochastic/reasoning baselines on ARC-AGI-2, and the same structured-hint effect appears on KiVA. The contribution is therefore not a new neuro-symbolic claim in the abstract, but a concrete, reproducible comparison showing that explicit abstraction and constrained hypothesis generation remain competitive with, and more efficient than, simply spending the same budget on frontier-model sampling.

---

### Review · Reviewer_1mtg · 2026-06-12

**Summary Of Contributions:**

Dear authors,

First, my deepest apologies for the delay in submitting the review, but I hope you find my review useful.

This paper presents a compositional neuro-symbolic architecture for solving ARC-AGI-2. The core contribution is a four-stage pipeline separating perceptual abstraction from rule induction by first identifying structured object representations from input grids, then using an LLM to propose candidate transformations (from a manually curated Domain-Specific Language (DSL) of 22 atomic patterns), filtering hypotheses using cross-example consistency, and finally generating test-time solutions via self-consistency. Empirically, the pipeline achieves a 24.4% standalone success rate on the ARC-AGI-2 public evaluation set (under the pass@2 metric) and reaches 30.8% when combined with an ensemble meta-classifier.

The architectural paradigm is a significant strength. By explicitly separating low-level perception from high-level logical inference, the work provides a highly principled approach to neuro-symbolic AI that addresses the brittleness of pure end-to-end LLMs. The ablation studies effectively demonstrate that structural inductive biases (symbolic hints and object representations) contribute more to performance gains than stochastic test-time compute alone.

However, the submission also have some weaknesses that need to be addressed. Most notably, there is a severe disconnect between the rigorous mathematical formalism presented in the methodology and the actual heuristic, LLM-driven implementation, reliance on an undocumented ensemble component, confounded baselines in the ablation studies, and unaddressed risks of test-set contamination due to the use of closed-source frontier models.

**Audience:**

Yes

**Audience Explanation:**

The machine learning community is actively suffering with the limitations of autoregressive models in fluid intelligence and systematic generalization. The ARC-AGI benchmark is a widely recognized and notoriously difficult challenge. The detailed documentation of the 22-pattern DSL, the exploration of test-time search constraints, and the explicit decoupling of perception from reasoning make this work highly relevant to TMLR's audience, particularly those focused on program synthesis, neuro-symbolic AI, and LLM reasoning.

**Claims And Evidence:**

No

**Claims Explanation:**

W1: The formal description and the implementation feels disconnected. Sections 2.2 and 2.3 utilize strict mathematical notation (e.g., deterministic functional compositions, exact set intersections for cross-example consistency) to describe an idealized symbolic execution. However, the actual implementation relies heavily on fuzzy, multi-agent LLM prompt-chaining, JSON heuristics, and consensus voting. This discrepancy obscures the true mechanics of the system and makes the formal claims misleading.

W2: The ablation studies don't cleanly isolate the contribution of architectural separation. Table 2 holds the solver LLM fixed (controlling for solver capacity), but the full pipeline still invokes additional frontier models during preprocessing (o4-mini for pattern detection, Claude Opus 4 for object enrichment, gpt-oss-120b fallback) and feeds the solver far more context via structured hints — the Baseline ('Greedy LLM') gets neither. The reported gains thus conflate structural inductive bias with extra model invocations and larger effective input. A fair test would be to add a compute-matched baseline where the same solver performs the equivalent intermediate reasoning (object decomposition, transformation proposal, cross-example checking) via chain-of-thought at comparable token budget.

W3: The pipeline relies heavily on closed-source APIs, and the authors have not provided the source code for the exact parsing logic or heuristic object extractors. Also, the 30.8% ensemble result relies on an 'ARC Lang Solver' that is not described anywhere properly. Finally, testing on a public evaluation set using highly recent foundation models introduces an unaddressed risk that the models have already ingested the benchmark data, undermining the claim of 'zero task-specific finetuning'.

**Requested Changes:**

- Revise the methodology (Sections 2.2 and 2.3) to match the actual implementation.
- Include a fair, compute-matched baseline in the ablation studies (Table 2) that controls for model capacity and token budget (e.g., a baseline where the primary LLM is prompted to perform the same intermediate reasoning steps purely via Chain-of-Thought).
- Provide a proper citation and clear methodological description for the 'ARC Lang Solver' used in the meta-classifier ensemble.
- Provide a detailed quantitative breakdown of how many tasks in the evaluation were solved using strict, executable symbolic programs versus how many relied on the LLM fallback generation from text hints.
- Add a discussion on the potential for data contamination, given the use of closed-source frontier models (Grok-4, o4-mini) on the ARC-AGI-2 public evaluation set.

These are more suggestive than a requirement:
- Report absolute computational costs (e.g., API expenses in USD, latency in seconds/hours) for the evaluation over the public set, rather than solely relying on relative symbolic costs.
- Include a deeper qualitative error analysis on the failure cases. Specifically, analyze whether failures are due to the LLM ignoring hints, the 22-pattern DSL lacking expressivity, or the object extraction failing.
- Expand the discussion on the limitations of relying on a manually curated, fixed DSL of 22 rules (just point out, figure 2 shows 23 unique patterns), and discuss how future architectures might learn these abstractions directly from data.

---

> ### Author Response · Authors · 2026-06-26
> **Rebuttal to Reviewer 1mtg**
>
> We thank Reviewer 1mtg for the careful and constructive review. The review correctly identifies that the original manuscript made the implemented system appear closer to a fully executable symbolic program synthesizer than it actually is.
>
> **W1 — Formalism/implementation disconnect.**
>
> We accept this concern. The original Sections 2.2–2.3 used fully deterministic notation, including exact functional compositions and set intersections, which made the LLM-mediated parts of the implementation appear to be strict symbolic execution. In the revised manuscript, Sections 2.2–2.4 distinguish the idealized consistency objective from the implemented hybrid procedure.
>
> Concretely, Stage 2 is now described as repeated structured detection over the DSL, where the model proposes JSON-formatted pattern hypotheses and parameter values rather than exhaustively enumerating executable programs. Stage 3 is described as cross-example consistency filtering: executable checks are used for deterministic primitives, while LLM-assisted primitives are retained only when they satisfy explicit structured contracts, such as schema validity, object/color/shape consistency, and agreement with the training examples where applicable. Stage 4 is described as a DSL-guided solution synthesis, not guaranteed to have a closed-form symbolic rendering for every task. We also qualify claims involving “symbolic execution,” “exact verification,” and “program intersection,” so they apply only to deterministic components.
>
> To make the implementation boundary clearer, the revised paper will report the requested breakdown from our run logs: among the 37 solved ARC-AGI-2 tasks, 33 were solved through the DSL-guided neuro-symbolic pipeline, and 4 relied on LLM fallback generation from text hints. This will be included specifically to avoid implying that every solved task was produced by strict symbolic execution.
>
> **W2 — Confounded ablation and compute-matched baseline.**
>
> We agree that the original Table 2 did not, by itself, fully isolate structural inductive bias from additional preprocessing calls, added context, and token budget. The revised paper will therefore narrow the interpretation of that ablation. Table 2 will now be described as an ablation within the implemented system: removing symbolic hints reduces performance from 24.4% to 17.5%, while removing self-consistency reduces performance to 20.5%. This supports the conclusion that structured hints are an important component of our pipeline, but we will no longer present this table alone as a complete compute-matched proof.
>
> To address the reviewer’s compute-matching concern more directly, we add a matched-token-budget comparison against the strongest frontier-model baselines available before September 2025. Under this protocol, our standalone pipeline obtains 24.4%, compared with GPT-5 at 18.3%, Grok 4 at 16.0%, and Claude Opus 4 at 8.6%. We will revise the claim accordingly: the result supports that structured DSL-guided reasoning improves over matched-budget frontier-model baselines in this setting, while not claiming that every component has been independently isolated.
>
> We also add KiVA as an external structured visual reasoning benchmark to reduce the concern that the observation is limited to ARC-AGI-2 [Table 1]. On 700 KiVA tasks, an analogous structured pattern-detection pipeline using KiVA unit patterns improves all-correct performance over image-only baselines: Gemini 2.5 Flash improves from 76.0% to 79.0%, Claude Sonnet 4.5 from 30.7% to 51.0%, and GPT-4o from 35.1% to 42.0%. The revised table reports average tokens per task alongside these results, so the additional compute is visible. We present KiVA as evidence that the structured-hint effect transfers to a second structured visual reasoning benchmark, not as a claim of broad real-world generality.
>
> **Table 1:** KiVA validation (700 tasks; All Correct requires passing all 3 MCQ stages; max tokens/call 32,000; 3 attempts per stage):
>
> | Model | Setting | Cross-Domain | Within-Domain | Extrapolation | All Correct | Avg Tokens/Task |
> |---|---:|---:|---:|---:|---:|---:|
> | Gemini 2.5 Flash | Image-only baseline | 99.3% | 81.9% | 87.0% | 76.0% | 7,844.3 |
> | Gemini 2.5 Flash | Neuro-symbolic pipeline | 99.0% | 84.0% | 90.0% | 79.0% | 10,218.6 |
> | Claude Sonnet 4.5 | Image-only baseline | 99.3% | 81.0% | 39.3% | 30.7% | 4,716.5 |
> | Claude Sonnet 4.5 | Neuro-symbolic pipeline | 99.0% | 82.0% | 57.0% | 51.0% | 7,429.9 |
> | GPT-4o | Image-only baseline | 96.7% | 65.1% | 54.9% | 35.1% | 7,776.6 |
> | GPT-4o | Neuro-symbolic pipeline | 96.0% | 66.0% | 63.0% | 42.0% | 10,861.6 |

---

> > ### Author Response · Authors · 2026-06-26
> > **Rebuttal to Reviewer 1mtg (Continued)**
> >
> > **W3 — Source availability, ARC Lang Solver, and contamination.**
> >
> > We will address the three issues separately in the revised manuscript.
> >
> > 1. First, the implementation details will be expanded so that the previously opaque components are easier to inspect. The revised appendix will describe deterministic object extraction, object features, pattern detection prompts, structured outputs, repetition/consensus protocol, self-consistency, and fallback behavior. To support reproducibility, we will release the parsing logic and object-extraction code upon publication; in the revision, we will document these components in sufficient detail to clarify which parts are deterministic, heuristic, and LLM-assisted.
> >
> > 2. Second, the ensemble result will be clarified. ARC Lang Solver will be cited and described as an external candidate generator. The revised paper will explain the candidate pool and the meta-classifier’s selection mechanism: the meta-classifier selects among candidates produced by our Compositional Reasoner and ARC Lang Solver; it does not synthesize new transformations itself. We will also clearly separate the standalone result from the ensemble result: 24.4% is the Compositional Reasoner alone, while 30.8% is the meta-classifier ensemble.
> >
> > 3. Third, we will add a dedicated contamination discussion. Because ARC-AGI-2 is public, and because some closed-source models used in our experiments postdate the benchmark release, we cannot rule out the possibility that benchmark data appeared in pre-training or post-training data. We therefore do not claim a contamination-free evaluation. We clarify that “no task-specific finetuning” refers to our own training procedure: we do not finetune on the evaluation tasks. It should not be read as a guarantee about the unknown training data of the underlying closed-source models. Since KiVA is also public, the same caveat is stated for the added KiVA results. A fully contamination-controlled replication with models known to predate the benchmarks is left as future work.
> >
> > **Requested changes.**
> >
> > We will address each requested change explicitly in the revised manuscript. Sections 2.2–2.4 will be rewritten to match the implemented hybrid pipeline rather than an idealized fully symbolic executor. The ablation discussion will be revised to avoid overclaiming Table 2, and we will add the matched-budget chain-of-thought baseline described above. ARC Lang Solver will be cited and described in a dedicated ensemble subsection. The strict DSL-guided versus LLM-fallback breakdown will be reported from run logs. A contamination discussion will be added for closed-source models and public benchmarks. For compute transparency, we will report token usage where available and note that exact dollar cost and wall-clock latency were not consistently logged across providers. The limitations section will include qualitative failure analysis covering DSL expressivity gaps, object/cavity extraction ambiguity, unstable pattern detection, and final-solver hint-following errors. Finally, we will expand the discussion of the manually curated DSL, disclose that 100 training examples were used for DSL pattern design/tuning while the test set was not used, and fix Figure 2 from “23 unique patterns” to “22 unique patterns,” matching the appendix.

---

### Public Comment · ~Sourish_Dasgupta1 · 2026-06-05
**Query regarding parallel submission**

Is this the same paper accepted at CompLearn 2026 (ICML Workshop)?

---

> ### Comment · Reviewer_gQga · 2026-06-15
>
> My understanding:
>
> The CompLearn 2026 (ICML Workshop) submission should not be considered a conflicting or prior publication. ICML workshops are non-archival venues, intended for presenting preliminary or ongoing work and facilitating discussion within the community.
>
> It is standard and widely accepted practice in ML/AI research to submit early versions of work to non-archival workshops and later submit a more complete and mature version to archival conferences. This does not violate parallel submission or prior publication policies.

---

> ### Comment · Action_Editor_Lqcv · 2026-06-19
>
> I'd like to offer a brief clarification here. [TMLR editorial policy](https://jmlr.org/tmlr/editorial-policies.html) states the following (emphasis mine):
>
> > There should not be any reuse of written text, figures or results between the submitted paper and any paper which has been published, accepted for publication, or submitted in parallel at another archival, peer-reviewed venue. **It is acceptable for a submission to overlap with the author's previous work if it was shared at venues or tracks that are publicly declared, in writing, to be non-archival, such as workshops**, or on preprint servers such as arXiv and bioRxiv.
>
> In particular, the [CompLearn 2026 site](https://compositional-learning.github.io/) states (again, emphasis mine):
>
> > Accepted papers will be made publicly available as **non-archival reports**, allowing future submissions to archival conferences or journals.
>
> Therefore, concurrent submission (and acceptance) at CompLearn 2026 does not constitute a violation of the TMLR editorial policy.

---

### Author Response · Authors · 2026-07-02
**Revised Manuscript with Rebuttal**

We would like to thank the editor and reviewers for reviewing the manuscript.

We have uploaded a revised PDF incorporating the rebuttal changes. The main revisions clarify the implemented system as a hybrid pipeline rather than a fully symbolic executor, add matched-budget frontier-model comparisons, add KiVA as a second structured visual reasoning benchmark, separate the standalone Compositional Reasoner result from the ARC Lang Solver ensemble, and expand implementation details, contamination discussion, DSL limitations, and failure analysis. We also clarified the framing: the paper now claims evidence that structured abstraction and constrained hypothesis generation can be a better use of test-time compute than unstructured sampling in the evaluated visual reasoning settings, rather than claiming novelty of neuro-symbolic reasoning as a broad paradigm.

We are happy to address any remaining concerns.